# Visual aids in ambulatory clinical practice: Experiences, perceptions and needs of patients and healthcare professionals

Catherine Hafner[1]*, Julie Schneider[2], Mélinée Schindler[3], Olivia Braillard[3]

1 Department of Endocrinology, Diabetology, Nutrition and Therapeutic Education, Geneva University Hospitals, Geneva, Switzerland, 2 Altitude 436 –Atelier graphique, Geneva, Switzerland, 3 Department of Community Medicine, Primary and Emergency Care, Geneva University Hospitals, Geneva, Switzerland

* Catherine.hafner@outlook.com

**Data Availability Statement:** Data are available: http://doi.org/10.5255/UKDA-SN-855201.

**Funding:** The funders (Edmond J. Safra Foundation) had no role in study design, data

## Abstract

This study aims to explore how visual aids (VA) are used in ambulatory medical practice. Our research group (two doctors, one graphic designer and one sociologist) have led a qualitative study based on Focus Groups. A semi-structured guide and examples of VA were used to stimulate discussions. Participants were healthcare professionals (HP) working in ambulatory practice in Geneva and French-speaking outpatients. After inductive thematic analysis, the coding process was analyzed and modified to eventually reach consensus. Six focus groups gathered twenty-one HP and fifteen patients. Our study underlines the variety of purposes of use of VA and the different contexts of use allowing the distinction between "stand-alone" VA used out of consultation by patients alone and "interactive" VA used during a consultation enriched by the interaction between HP and patients. HP described that VA can take the form of useful tools for education and communication during consultation. They have questioned the quality of available VA and complained about restricted access to them. Patients expressed concern about the impact of VA on the interaction with HP. Participants agreed on the beneficial role of VA to supplement verbal explanation and text. Our study emphasizes the need to classify available VA, guarantee their quality, facilitate their access and deliver pertinent instructions for use.

## Introduction

Images are used to communicate and teach in a multitude of areas. In the medical field, the recall of information transmitted orally during consultation is unsatisfactory and written documents are often not adapted to patients [1]. The use of images seems to improve understanding, attention and recall of information [2]. A lot of interest has been devoted to the development and use of pictograms to improve medication adherence [3–5].

For the purpose of this article we have defined "visual aids" as all media or formats that are used to give information with the aid of non-moving images (aid to verbal or written information). Images comprise photographs, illustrations, drawings, infographics (method to visually communicate information) or pictograms (Fig 1) [6].

collection and analysis, decision to publish, or preparation of the manuscript. Funds were used to pay for salary for one person to transcribe verbatims and fees for one author (JS), as well as for various costs related to the organization of the study. Three of the authors are employed by HUG, while one author: Julie Schneider is an independent graphic designer (at Altitude 436), her time producing materials for the study was financially compensated, without any conflict of interest. The funder provided support in the form of salaries for authors JULIE SCHNEIDER, but did not have any additional role in the study design, data collection and analysis, decision to publish, or preparation of the manuscript. The specific roles of these authors are articulated in the 'author contributions' section.

**Competing interests:** The authors have declared that no competing interests exist.

Other types of VA than pictograms (illustrations, infographics, photo stories and cartoons) have also been studied for patient education and risk communication among other fields [7–14]. Studies focus on a single type VA, a single purpose and take place in precise research context [15, 16]. It is therefore complicated to understand how visual materials can take place into real daily practice with the currently available data.

While many studies focus on the process of creation and validation of visuals, this study aims to examine the way users can profit from visuals already available. We want to identify the different types of VA that are used in clinical practice without focusing on a particular visual element or a specific purpose. Our aim is to understand how they are used by HP and patients and to identify benefits and constraints in real practice. The ultimate goal is to facilitate the use of VA in clinical practice in a safe and relevant manner.

# Method

## Context

Our study took place at the at the Geneva University Hospitals (HUG) which dispenses ambulatory care of general internal medicine to a population of different socio-educational backgrounds and diverse nationalities and cultures. Geneva is undeniably a cosmopolitan canton whose proportion of foreign nationals in the population is the highest in Switzerland [17]. This illustrates the variety of nationalities, cultures, spoken languages, socio-educational and economic levels in our canton and the potential communication hurdles that can ensue.

## Methodological approach

**Design.** We conducted a qualitative study based on focus groups. Following a review of the literature, a guide to a semi-structured focus group (FG) was established in order to stimulate discussion. The interview guide (see Table 1) was validated in a pilot FG. Given the absence of modification to the guide, the pilot FG was included in the analysis.

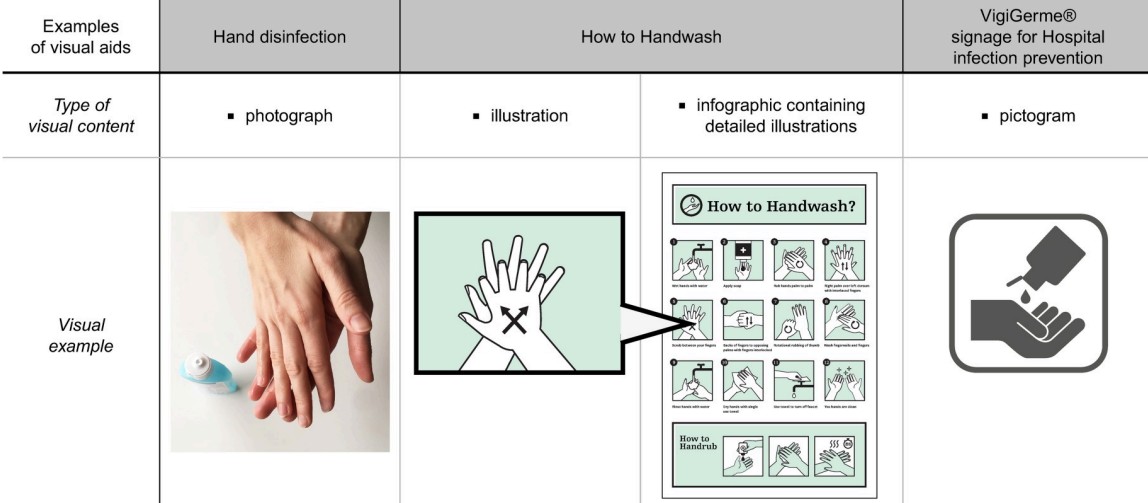

**Fig 1. Examples of different types of images to represent the "hand disinfection" concept, using a photograph, an illustration, an infographic with illustration and a pictogram.** "How to Handwash?" under a CC BY 4.0 license, with permission from Altitude436, original copyright © Altitude436, 2021. Pictogram reprinted from "Vigigerme® Hygiène des mains" under a CC BY license, with permission from HUG, original copyright, © HUG 2017.

**Table 1. Interview guide used in the focus groups.**

| |
|---|
| (1) **Explore perceptions of participants related to visual aids (VA)** |
| • Could you give us examples of VA? Examples in the medical field? Who are they for? |
| (2) **Determine acceptance criteria for VA by exploring the views of patients and HP regarding VA as well as collecting their reactions to different types of images.** |
| • Could you comment on this selection of images? (Presentation by the moderator of a selection of VA related to the medical field. The infographics were chosen to represent a palette of diverse tones and aesthetics: humorous, scientific and symbolic versus descriptive etc. Only paper-based VA were presented.) <br> • Do you enjoy using images to transmit information? Why? <br> • Could you give us personal examples where you have been confronted with images related to the medical field? <br> • Do you perceive that the use of images helps the transmission of information? How? Why? |
| (3) **Identify determining factors that would influence the use of VA, in particular by questioning participants on the areas of healthcare where VA would be perceived as being the most useful and by exploring the needs and expectations of patients and HP.** |
| • What criteria must be applied to a visual aid for it to be useful? <br> • Do aesthetics play a role in the interpretation of an image? <br> • What influences the use of VA? <br> • How could images be integrated into daily care? <br> • What areas would be the most relevant? (Examples: treatment, prevention, explanation of diseases, explanation of the healthcare system, anatomy). <br> • Which format seems to you the most suitable? Paper versus digital. |

**Research team.**    The research team included two general practitioners, a sociology researcher and a graphic designer. One of the doctors and the sociology researcher are experienced in qualitative research and in patient education.

**Conduct.**    The research protocol was sent to the Research Ethics Council, which confirmed that in light of our methodology there was no need for a full submission. The inclusion criterion for all participants consisted of oral proficiency in French. For HP, the criterion of inclusion was healthcare practice in Geneva with ambulatory patients without focusing on a specific profession. Regarding patients, inclusion criteria consisted of having an experience of ambulatory medical follow-up in Geneva and not being hospitalized.

We carried out a purposive sampling in order to have the widest range of HP and patients. Criteria used for purposive sampling of patients were age (> 2 patients over 70 years old), native language (> 35% of non-french native language) and number of past ambulatory consultations. We aimed to have different types of ambulatory experiences (chronic disease follow up, emergency consultation, preventive care e.g.). Due to the impossibility to include non-french speaking patients in the FG, we chose the native language criteria to ensure a diversity of cultural backgrounds. The recruitment of HP was done via e-mail and posters. The pharmacists were recruited by e-mail and visits to dispensaries closest to the hospital in order to avoid too large recruitment.

The patients were recruited in waiting rooms or through posters in the hospital and in city pharmacies as well as directly by doctors working in the HUG. All participants received a gift voucher of CHF 20.—for a department store. Lunch was offered during focus groups and each participant signed informed consent.

Focus groups were moderated by the sociology researcher and took place between 24.05.2017 and 26.07.2017. One to two other members of the research group took handwritten notes of the sessions in addition to the audio recording. We have separated patients and HP for the focus groups in order to facilitate discussion within the group. The recorded interviews were transcribed ad verbatim by a remunerated external person. A proofreading was performed in order to verify the accuracy of the transcript.

## Data analysis

All the members of the research group read the transcripts and established by consensus a first code list in an inductive manner with a precise definition of each code, so that the coders could refer to it during the analysis. There was no disagreement requiring an intervention of a third party.

Two members of the group have encoded the full content of the discussions using an analysis program (*Dedoose*, Version 8.1.21). The totality of the content was double-coded. The list of codes was refined as coding continued, always with the consensus of the entire research team.

## Results

Three "healthcare professionals' *focus groups*" (including the pilot FG) and three "patients *focus groups*" achieved data saturation. Fifteen patients and twenty-one HP participated. Patients groups lasted an hour and a half, HP sessions one hour. Table 2 enumerates specific occupations of HP and Table 3 patients' characteristics. The 3 focus groups with HP included minimum 4 different professions.

The inductive thematic analysis resulted in bringing out five main themes: 1) identification and definition of VA 2) context of use, purpose and role of VA, 3) co-construction and reference systems, 4) accessibility and quality of VA and 5) patient-healthcare professional relationship.

**Table 2. Healthcare professionals.**

| Occupation | Number |
|---|---|
| Dispensary pharmacists | 4 |
| Physicians | 4 |
| Nurses | 7 |
| Dietician | 1 |
| Physiotherapist | 1 |
| Occupational therapists | 2 |
| Radiology technicians | 2 |

**Table 3. Patients' characteristics.**

| Characteristics | Number |
|---|---|
| Age: | |
| <40 years old | 3 |
| 40–70 years old | 8 |
| > 70 years old | 4 |
| Native Language | |
| French | 9 |
| Other | 6 |
| Number of past ambulatory Consultation | |
| <5 | 6 |
| 5–50 | 6 |
| >50 | 3 |

## Identification and definition of visual aids

When asked about examples of VA, the HP listed a multitude of visuals. Complete elaborate VA (infographics, posters, illustrated sets of cards) were cited as well as individual images such as pictograms or illustrations. Drawings, photographs and videos have also been mentioned as examples. The paper format is often used but its source is frequently digital. Original variations were also cited by patients when asked about VA such as the use of stones.

> Patient: "I have a dermatologist who has stones at his medical office that are laid one on top of the other and he told me that's the skin. So that's the dermis, the epidermis, he showed me all the layers, but with a name." (FG6)

While patients mostly described positive experiences with VA, a few expressed that a good explanation was enough.

> Patient: "I think, a good doctor, or a good female doctor, explains it to you and for me it is enough. I don't need images." (FG6)

In the end, examples of VA listed by participants include a great variety of types and formats. The users do not seem to identify different categories of visual elements (pictograms versus illustration for example) but they focus rather on the roles and purposes of VA.

## Contexts of use, purposes and roles of visual aids

We have identified two main contexts of use of VA. On the one hand, some VA were mainly created for use outside consultation, either by patients or HP on their own (posters, brochure, medical file etc.). We propose to call this type of VA: "stand-alone VA". On the other hand, the majority of examples given by participants concerned the usage during consultation where a visual aid is presented/created by the healthcare professional to interact with the patient. We propose to name those VA: "interactive VA".

Participants cited a few examples where VA were used out of consultation, e.g. posters in waiting rooms or hospital orientation signs. Their primary purpose was unilateral transmission of information or spatial orientation. These "stand-alone VA" as opposed to "interactive VA", are used without healthcare professional's guidance. Participants underlined that in order to transmit information, they rely considerably on the text, given the absence of verbal explanation. However, if the stand-alone visual aid is a reminder of a known information, the image alone can be very helpful.

> Healthcare professional: "Afterwards, I think, these are things that are clear to us because we see them every day. But afterward, if you are a patient, it will probably, be the first time that you will be confronted with this image. There we will have to explain to him at one time or another otherwise we risk not being on the same wavelength. . . precisely that risks causing problems later. . .we start on a bad note." (FG1)

For the majority of patients, the primary role of images in the context of stand-alone VA is to attract attention.

> Patient 1: "There is too much text in relation to the image. And what must strike is the image and not the text, that's what strikes people."

> Patient 2: "Yes sometimes it's a strong and simple image" (FG5)

This role seems important given that compared to text alone, participants were clearly more attracted to infographics combining both images and text.

Moderator: "And if there had been the text alone, you would have read it too or not?"

Patient: "If there is too much reading, no. Because, finally, there are sometimes terms in the things you are given, therapeutic names, you really have to go back and ask again later to find out what it means (. . ..)." (FG4)

However, participants underlined the risk of overloading stand-alone VA with details or a multitude of information because essential meaning could be missed.

Moderator: "Yes but why more speaking?"

Patient: "Because there are less images, one can more easily catch the. . .. because the others, it is a bit, messy, one can't. . ..one doesn't know where to look." (FG 4)

Participants also emphasized the importance of the "tone" given to the message. Indeed, an image can also discourage the patient as exemplified by participants' reactions (1 patient and 2 HP) to an infographic about diabetes with a seemingly frightening pictogram showing an amputated person.

Healthcare professional 1: "Well, it needs to catch attention, but I mean. . . Aggressive at this point. . ."

Healthcare professional 2: "Yeah. . . not that it's too aggressive either but. . .all the same you understand right away, it has to be understood right away."

Healthcare professional 3: "I think it's hard to do both. . . that there is at the same time a reaction from the general public but also that it doesn't scare too much, in order to de-stigmatize the disease a little, finally. As we do for AIDS-HIV." (FG1)

Another cited type of stand-alone visual aid consists of pictograms for Hospital Hygiene Signage (Vigigerme®) [18]. Those pictograms were listed as examples of efficient stand-alone VA which render the transmission of information easy and became so common that patients and HP often do not need additional verbal or written information.

Regarding the use of VA during medical consultation as "interactive VA", different purposes were identified: explanation, overcoming communication barrier, recall and situation setting. Table 4 allows to list some examples of "interactive VA", as opposed to "stand-alone VA", also with mention of roles, purposes, advantages and constraints.

In the context of a medical consultation, VA are often used as tools to improve communication when the circumstances are challenging (linguistic barrier, aphasia, illiteracy, hypoacusis). Some HP underlined the lack of interactive VA for translation purposes when oral communication is impossible and requested tools designed for translation. Others, on the contrary, underlined their determination to avoid substitution of oral communication with interactive VA because of the risk of misinterpretation.

Healthcare professional: "I find that [. . .] finally, the visual tools are more. . . finally, useful for the patient, after, finally I find it really necessary to have, finally either to construct them with the patient, or that there is an explanation provided at the same time as the presentation of these tools." (FG2)

**Table 4. Purposes and roles of interactive visual aids with cited examples (not exhaustive) and their advantages and constraints of use.**

| Purpose | Examples (not exhaustive) | Role of visual | Advantages and constraints of use |
|---|---|---|---|
| Explanation | Anatomic sheets | Helps to explain information that depends on the visual. | Level of complexity can be adapted. |
| | Drawing of a surgical procedure | | |
| | | | Helps with verification of understanding |
| | Visual representation of risk | Helps to understand complex information. | Possibility of making the patient active. |
| Overcoming a communication barrier | Photograph depicting acts of daily life | Support for communication | Risk of use as a translation tool for language |
| Recall | Medication schedule | Simplifies information. | Possibility of co-construction with patient. |
| | Dietary sheets | Decreases the amount of text. | Allows dialogue. |
| | | Draws attention. | Helps with verification of understanding. |
| | | | Can impact motivation. |
| Situation setting | Images of acts of daily life | Helps to project oneself | Possibility of exploration, verification of understanding Roleplay |
| | | Abstraction | |

Even when communication is not an issue, interactive VA are used to help with explanation of visual information, especially related to anatomy, or complex concepts (statistical risk e.g.) by representing them in an illustrated way.

Patient: "I also have a small ear problem, and then. . . often you can't actually picture the problems one has, and then by drawing, sometimes it can be of help. I think it's simpler, it's more understandable." (FG 4)

HP can also benefit from the use of a visual as pointed out by this patient:

Patient: "I think you put this in front of the doctor, who will explain this in front of the patient; well yes, it is much easier, for the doctor, for the explanation and at the same time for the patient to understand." (FG 5)

Indeed, HP report that using an interactive visual aid is an opportunity to simplify information and therefore to select the main message to be transmitted.

Healthcare professional: "there is [. . .] so much info now, between the Internet, applications, in any case regarding food, and I tell myself that going back to things a little bit essential, it can also help to sort info, even for professionals, for us, and then for the patients." (FG 1)

Another purpose of use is to help with the recall of information. The interactive visual aid used during consultation can become a reminder. Indeed, participants (patients and HP) particularly cited interactive VA using a format which can be consulted at home.

Patient: "we can take it with us, it's clear. And it was very useful to me precisely because she [his wife] was not there and when she arrived, I explained to her, I had to pull out the graph. Yes, I pulled out the graph, and voilà, really, I understand even more." (FG 4)

Situation setting is also a purpose for the use of interactive VA when images represent situations of daily life or a future intervention. Their role is to help the patient to project himself into a different context in order to anticipate potential difficulties and answer questions.

The main common characteristic of interactive VA is how they are integrated during consultation by the healthcare professional allowing the patient to react, facilitating discussion and interaction. It can help patients to express their choices and the healthcare professional can verify the patient's understanding.

Healthcare professional: "but even just having a little visual to have the patient confirm, whether he understood or not, is good. Whether by explanation or a drawing, he should say yes I have really understood, or not I have not understood, so that we can change the technique, in case he did not understand." (FG 3)

The patients agreed that they also wanted to take an active role, to guide the healthcare practitioner and to create a dialogue while using interactive VA.
Patient 1: "Yes absolutely, yes, we share it at the same time."

Moderator: "If you, you understand, he goes further, or he goes back to. . ."

Patient 1: "See, the intestine, it's that and that, and then etc. and then you show it all together, if you want." (FG4)

Healthcare professional: "If I can add something, a good tool is necessarily interactive, so a ready-made thing is not very effective, so the patient must have something to do, whatever the activity, but that he have something to do." (FG 2)

The active role of the patient while using interactive VA is essential but it should also be the case in the process of creation of VA.

## Co-construction and reference systems

HP identify the risk of using overelaborate symbols, not adapted to the health literacy level-education level or symbols based on an occidental cultural reference not shared by patients.

Healthcare professional: "we tend to put our images, our perceptions, or those which would correspond to our culture, and often they can be misinterpreted. I always have in mind one case in South America where in fact an alcoholic beverage was marketed with the pictogram of a pregnant woman, who should not drink, but then the population understood that if you are not pregnant, you cannot drink this beverage." (FG 2)

A telling example is that of the medication schedule card.

Healthcare professional: "one quickly realizes that even a simple double entry table with the name of the drug on the left and schedules on top, most people don't understand, because it's a concept, it's a thing we learned in school, which to us seems to be absolutely obvious, but which is not at all obvious to some people, it still remains something not simple and finally what they understand best, that's right." (FG 3)

Indeed, during focus groups we have not been able to identify a systematic pattern of how VA are read by patients. Some patients focused on the text, others on details of the images or the atmosphere of the VA. We also found that a too literal interpretation of images can be confusing, for example on dietary sheets representing meals where patients literally focus on the illustrated food instead of the concept of a meal: breakfast, lunch, dinner, etc.

Patient: "for me, compared to that one, it took me a little time to understand, because I know what it is about, [. . .] because I am dieting, I know what it is about. But for someone who has normal habits [. . .], he may think that it is an advertisement for oil." (FG 5)

Two solutions were suggested in order to prevent the risk of confusion related to a discrepancy in reference systems of the healthcare professional and the patient: the co-construction of VA with patients was proposed by HP, and the acquisition of a common reference system by both HP and patients. The medication schedule card for example can be co-constructed with the patient. The often cited example of traffic signs supports our ability to develop and learn a common language thanks to their high visibility in our daily lives.

Healthcare professional: "I think that before SMS existed we all had the reference of the Highway Code. Since then, emoticons have become widely spread. So now do we have to create new ones or should we be inspired by them?" (FG 1)

## Accessibility and quality of visual aids

Given the notion that patients and HP could learn a common reference system, participants emphasize the importance of sharing VA, enabling the construction of a common visual language. Some HP are also interested in sharing access to VA with patients.

Healthcare professional: "[. . .] I also think what is interesting is that the documents could be shared, [. . .] visible both from the outside by the patients and also by us, [. . .] it can already nourish [. . .] consultations and also that they can already have access to the documents, perhaps before coming to consultation so they have time to read a little. . ." (FG 1)

HP complain that the difficulty of accessing VA restricts their use. Therefore, participants suggest the creation of a digital database.

Nevertheless, HP mention that the quality of VA should be guaranteed and the source identifiable. Some HP qualify as "Do It Yourself" the creation of VA by themselves or doing a Google search in front of the patient. HP are concerned about an inappropriate message or a misunderstanding if VA are not validated.

## Patient-healthcare professional relationship

Patients seem to appreciate any effort aimed at conveying clear information to them. They appreciate being adequately informed and often report situations where medical jargon or the complexity of the information limits their understanding. The fact that an image is drawn or explained by the healthcare professional seems to add value to the tool.

Patient: "I think a drawing by a doctor, I don't ask that he is fully professional or a graphic designer, but that he explains to you with a drawing and that he explains it to you at the same time that he does it, then it's meaningful [. . .]. Yes, precisely very meaningful because my doctor did this for me." (FG 4)

Thus, any effort to convey clear information, whether it takes the form of a visual aid or not, is appreciated and probably strengthens the relationship.

We have observed that patients are concerned about the time of interaction with the healthcare professional.

Patient: "Because it takes work on the part of the doctors. And when I see that they barely look at the person. . .they're stressed out there because there's the other one waiting (. . .). At the start, well he gives an explanation (verbal) (. . .) but then I don't know if we can still look at the drawing, there indeed. . . the time. . ." (FG 5)

Any tool which may end up as a substitute for this exchange with a healthcare professional would constitute a threat to the patient-healthcare professional relationship. However, VA which synthesize information or focus on key points are cited as beneficial for recall because HP sometimes do not have time to summarize or repeat the main points at the end of an interview.

## Discussion and conclusion

### Summary of results

HP described a variety of contexts for the use of VA with different roles, formats and sources. They question the quality of certain VA and seem restricted by the lack of easy access to referenced tools. Co-construction of VA is a significant asset to them.

Patients experienced difficulty identifying what VA are and how they are useful. Concern was expressed that they will replace the interaction with HP.

We have found that VA are highly appreciated when they replace or supplement written information without being a substitute for oral explanation. Patients appreciate that HP transmit quality information to them but the challenge is the synthesis and simplification of explanations.

### Strengths and limitations

The purposive sampling may have selected participants interested in VA and patient education. The broad scope of the VA and the single site methodology might limit the exhaustiveness of the results. As in any qualitative study, data collection and analysis may be subject to bias. However, our research team of various professions and career paths has brought wealth and weight to the qualitative analysis. This study is innovative by its transversal approach to various areas of care and the active participation of both patients and HP.

### Discussion

Our transversal approach revealed that participants are exposed to a wide diversity of VA (pictograms, illustrations, infographics, etc.). Both HP and patients do not distinguish VA by their visual type (pictograms vs illustration, e.g.), but rather by their purpose (information vs situation setting e.g.). Participants emphasize the importance of general aesthetics and the need to catch the patient's attention. This fact validates the importance of collaboration between patients, HP and graphic designers to take into account purposes as well as visual needs [2, 19, 20].

Our study underlines the difference between a stand-alone visual aid given out of consultation and an interactive visual aid used with a patient by a healthcare professional during consultation. Only the latter provides an opportunity to initiate dialogue and make the patient active. This finding is paramount to this study as most of the benefits and constraints of VA identified in this study are related on how the VA is used as an interactive visual aid (vs a stand-alone visual aid) rather than which VA or graphic type should be used.

The main risk of using VA described by both HP and patients is misinterpretation, especially when VA are used as a support out of consultation and without explanation or when the

patient has a different cultural background. Without distinction of context, recommendations on the use of VA agree on the importance of verbal and/or written explanation to accompany graphic elements in order to reduce the likelihood of misinterpretation [2, 21].

The literature has demonstrated the risks of communicating exclusively with the use of graphical symbols because they are not inherently universal [22–24]. A study proposes the creation of pictogram "sets" when translation is not possible but they must be adapted and validated for the target population [25]. Cultural background is essential to validate VA for a specific population. However, we were not able to define any specific pattern of reading patterns or preferences of patients of similar characteristics in our study, underlying the complexity of this topic and the need of further individualization of a VA.

Many participants (patients and HP) express the importance of being able to personalize VA and even co-construct them during consultation. The benefit of integrating the patient in the creation of visual media is mainly developed in the literature for the creation of visual databases for a specific population [20, 26]. In patient education, the co-construction of a pedagogical aid, whether in the process of creation or during consultation, is considered necessary to encourage patient engagement [21, 27]. Co-construction of an interactive visual aid is then not only a creation process with the possibility of individualization but it can also become a pedagogical process [20]. HP should however be trained for such an approach.

Although our findings emphasize benefits of VA when they are used as interactive VA rather than stand-alone VA, verbal communication around VA is not frequently assessed in research studies. The recommendations focus rather on the necessity of pointing to images while talking or explain the benefits of an image for a specific purpose as compared to another [2, 6, 28]. We think that the addition of instructions for use is thus necessary. This would ensure pertinent and safe usage and guide HP to promote dialogue and partnership with the use of interactive VA.

The HP state many constraints to the use of VA in their practice: complicated access and lack of guarantee of quality. These elements are consistent with the literature which underlines the importance of following through the processes of creation and validation of VA [2, 3]. The authors note the significant cost of this type of development and the lack of an accessible free database [16].

The concept of learning a visual language is addressed by HP. The creation of universal pictograms or pictograms personalized for each individual is unrealistic [16]. Research avenues are proposed in this direction to determine whether exposure to a repeatedly occurring visual language associated with a verbal explanation could make acquisition possible [22]. Given the limitations cited above, an interesting prospect would be the coordination of all efforts to create a common visual language and teach patients and HP to understand and use it, similarly to traffic signs.

## Conclusion

Current research highlights the strengths of integrating VA into medicine as well as the constraints. VA integrated in consultation can become valuable tools for explanation, fostering dialogue with the patient, supporting learning and cultivating the therapeutic relationship. The main constraints don't question the VA usefulness or efficacy but rather the processes that might endanger their use: lack of access, quality assessment or individualization to the patient need or preferences. Future research should shift from "which" standardized visual aid should be used to "how" to integrate them into daily practice.

## Practice implications

VA are useful in clinical practice for different purposes as information, overcoming communication barrier, recall or situation setting, but should not replace oral communication. VA

should be co-developed with patients, knowing that a unique VA can't be adequate for all patients. HP and patients should be able to personalize them in the way that VA supports patients to express their needs and preferences. The development of a common reference system might also be an alternative to ensure VA's adequacy with different profiles of patients. VA should also be easily accessible and their quality guaranteed to enable their use in daily practice.

## Author Contributions

**Conceptualization:** Catherine Hafner, Julie Schneider, Olivia Braillard.

**Data curation:** Catherine Hafner, Julie Schneider.

**Formal analysis:** Catherine Hafner, Julie Schneider, Mélinée Schindler, Olivia Braillard.

**Funding acquisition:** Catherine Hafner.

**Investigation:** Catherine Hafner, Julie Schneider, Mélinée Schindler, Olivia Braillard.

**Methodology:** Catherine Hafner, Julie Schneider, Olivia Braillard.

**Project administration:** Catherine Hafner.

**Supervision:** Olivia Braillard.

**Validation:** Catherine Hafner, Olivia Braillard.

**Writing – original draft:** Catherine Hafner, Julie Schneider.

**Writing – review & editing:** Mélinée Schindler, Olivia Braillard.

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
