## [Decision Letter · Decision Letter 0]

8 Dec 2020

PONE-D-20-29419

Visual aids in clinical practice: experiences, perceptions and needs of patients and healthcare professionals

PLOS ONE

Dear Dr. Hafner,

Thank you for submitting your manuscript to PLOS ONE. After careful consideration, we feel that it has merit but does not fully meet PLOS ONE’s publication criteria as it currently stands. Therefore, we invite you to submit a revised version of the manuscript that addresses all points raised during the review process.

We look forward to receiving your revised manuscript.

Kind regards,

Barbara Schouten

Academic Editor

PLOS ONE

2. Please provide additional details regarding participant consent. In the ethics statement in the Methods and online submission information, please ensure that you have specified (1) whether consent was suitably informed and (2) what type you obtained (for instance, written or verbal) and how this information was recorded. If your study included minors under age 18, state whether you obtained consent from parents or guardians. If the need for consent was waived by the ethics committee, please include this information. Please also state the full name of the IRB institute.

3. We note that Figures 1 and 2 in your submission contain copyrighted images. All PLOS content is published under the Creative Commons Attribution License (CC BY 4.0), which means that the manuscript, images, and Supporting Information files will be freely available online, and any third party is permitted to access, download, copy, distribute, and use these materials in any way, even commercially, with proper attribution. For more information, see our copyright guidelines: http://journals.plos.org/plosone/s/licenses-and-copyright.

(1) You may seek permission from the original copyright holder of Figures 1 and 2 to publish the content specifically under the CC BY 4.0 license.

4. Thank you for stating the following in the Financial Disclosure section:

"Catherine Hafner

CGR 75659

Edmond J. Safra Foundation.

https://www.edmondjsafra.org/

The funders had no role in study design, data collection and analysis, decision to publish, or preparation of the manuscript"

We note that one or more of the authors are employed by a commercial company: Atelier Julie Schneider Graphic Design, Carouge, Switzerland

(2) Please also provide an updated Competing Interests Statement declaring this commercial affiliation along with any other relevant declarations relating to employment, consultancy, patents, products in development, or marketed products, etc.  

Reviewers' comments:

Reviewer's Responses to Questions

**Comments to the Author**

1. Is the manuscript technically sound, and do the data support the conclusions?

Reviewer #1: Yes

Reviewer #2: No

2. Has the statistical analysis been performed appropriately and rigorously? 

Reviewer #1: N/A

Reviewer #2: N/A

3. Have the authors made all data underlying the findings in their manuscript fully available?

Reviewer #1: No

Reviewer #2: No

4. Is the manuscript presented in an intelligible fashion and written in standard English?

Reviewer #1: Yes

Reviewer #2: Yes

5. Review Comments to the Author

Reviewer #1: Many papers have been published about visual aids. This manuscript adds relevant additional knowledge to current insights based on qualitative interviews. The research question is well embedded in literature, the methodology fits, the results are described clearly. My main issue is that I would expect some more citations to illustrate the findings, as some paragraphs don’t use any citation at all. I have the following minor issues as suggestions for further improvement.

Methods:

• When was the study performed?

• Focusgroup: what number of participants was your aim? What were criteria for purposive selection (e.g. gender, age, health literacy, experience)?

• With respect to the criteria for purposive selection: did you find any indication how the results were influenced by these criteria? Did you look for conflicting evidence?

• The definition of VAs is somehow problematic, as you did leave this to the focus group (and later you give direction to the discussion by giving examples). Hence it is not clear what kind of VAs this research addresses.

o In general, I believe this is not problematic in the context of this paper. As the results show, different types of VA were named during the FGs.

o Please discuss this (also with respect to more modern/complex interventions that include visual material, such as instruction video’s, animations, instruction games).

o Did you notice any differences in benefits and limitations with respect to the type of VAs?

• Please include the interview guide in the manuscript (not as supplement).

Results

• The use of citation seems unbalances (no citation in the first paragraphs, appropriate citations in the second part)

• I believe line ‘117 Professions of healthcare participants’ can be skipped.

• Paragraph Identification.

o Please support your findings in this paragraph with appropriate citations.

• Paragraph context

o I believe Table 2 might be skipped as the information is well covered within the text.

o Illustrate the first part ‘context’ with citations

o Should the text below the Table be part of the Title of Table 3 ‘Purposes and roles of visual tools with cited examples (not exhaustive) and their advantages and constraints of use.’?

o Line 214, page 12. What is the active rol (as this is not clear)? Please give an example: The active role of the patient while using “visual tools” is essential but it should also be the case in the process of creation of VA.

• I believe the paragraphs Summary of results and Strengths and limitation should be part of the discussion

Discussion:

• Some more limitations might be discussed, such as the interview guide, the very broad scope of the topic (ranging from pictograms to games).

• Discusion please include the following aspects in your discusion

o How do the cultural background and patient characteristics such as age, gender and disease characteristics influence the need for VA, and what cultural aspects are most relevant in this context?

o Co creation: very interesting topic. Is this meant in the context of development of VA, or might this also apply to the use of VA during every single consultation?

o How should be accounted for diversity in health literacy in the use and development of VAs?

Remark:

• Statement concerning data availability: it seems that the underlying data (complete transcriptions, analysis and audio-taped focusgroup interviews) are not available. I do not believe this is needed for this paper. However I believe the statement that “all data underlying the findings described fully available, without restriction, and from the time of publication” is not true.

Reviewer #2: Overall impression

The authors describe a study consisting of focus groups with patients and healthcare professionals on the use of visual aids in ambulatory medical practice. Their aim is to understand how visual aids are used and to identify benefits and practical constraints. The authors conclude that there is a need for visual aids in medical practice, that patients are concerned about replacing the conversation with visual aids and that practitioners are looking for support with access to high quality materials. It is a topic worth exploring, with a commonly-used methodology and great to see a multidisciplinary researcher team. However, there are major weaknesses in the paper with regards to presenting the data (i.e., only for some part of the Results are data presented in a way that shows they follow from the interview; other sections read more like the authors' interpretations) and logic (i.e., conclusions are drawn that do not seem to follow from the presented data and that are not in line with the presented research questions). In addition, the practice implications to create a database with visual elements does not seem to follow from the study findings.

Introduction

1. The introduction should ideally begin with an explanation of the problem that will be addressed in the study, along the lines of “In the medical (…) adapted to patients”, lines 47-49

2. 41-42: The definition of visual aids appears to be incomplete. The description seems to cover ‘visual’ without focussing on the role of ‘aid’

3. The examples in lines 42-43 are incomplete and not very informative. There are several mentions of ‘.etc’ in the first few pages (lines 38, 42, 52) where it would be more appropriate to present a fuller context.

4. 52: the ‘therefore’ does not follow logically from the sentence before. The authors may want to consider adding something like “It is therefore difficult to draw evidence-based conclusions on what can be considered good practice…”

Methods

5. The described methods generally sound appropriate, but are not tailored to the question on insight into current use of visual aids, for which you would preferably have a wider cross-section. The described method would be better suited to address the question on stakeholders’ perceptions on helpful ways(/barriers) to use visual aids instead.

6. In addition to this, the focus group guide (supplementary file) does not appear to address practical constraints, which is one of the indicated research questions.

7. Line 99-100: Please expand on why the separation of patients and HCPs seemed important or how it helped facilitate discussion, e.g. did they feel more comfortable to share their stories?

8. Line 109: Please provide the intercoder reliability for the double-coded section.

Results

9. Major issue - This section does not meet standards for qualitative reporting. It is generally unclear what was discussed in the interviews and what is interpretation by the authors (e.g. line 214-215: “The active role … creation of VA”). It is also unclear whether findings were mentioned once by a single stakeholder or whether there was group consensus – it would be helpful to add numbers or general indicators (“the group agreed that..”). Quotes should be added, like in section 178-202. The summary of results (283-291) indicates the potential of interesting findings from the interview data, but this is not adequately presented in the results section.

10. It is great to see the diversity of healthcare professionals included. I am missing the characteristics/demographics of the patient participants. Characteristics such as being a chronic vs new patient, age, literacy levels and cultural background can have great impact on preferences for the use and look of visual aids.

11. 125: what do the authors mean with “complete elaborate VA”?

12. 129-135: this is more a methodological consideration re the researchers’ use of language in the interviews than a result to report.

13. 142+145: the authors make a distinction between ‘visual support’ and ‘visual tool’. However, the language used for this distinction is not very intuitive. If I understand the difference right, I would suggest using ‘stand-alone visual aids’ vs ‘interactive visual aids’. This is in line with what the authors state in line 206 that ‘the main common characteristic of visual tool is their context of use’.

14. 164: The authors list ‘some examples of visual support’ in table 2. This list needs to be extensive, as the aim of the work is to understand how visual aids are being used.

15. 166, Table 2 – also, 174, Table 3: Were the roles of the visuals and advantages and constraints of use discussed in the interviews, or are these the authors’ interpretations?

16. 166, Table 2: The body of the text talks about the importance of ‘tone’ and ‘frightening pictogram’. The table does not mention emotional effects as a consideration for use.

17. 174, Table 3: The authors may want to think about a more specific term for ‘Communication’ as a purpose, as explanation and situation setting are also forms of communication.

18. 185: Could the authors clarify what they mean with ‘typically visual information’?

19. 230: Could the authors clarify what they mean with “we have not been able to identify a system of how VA are read by patients”

20. 238: Who offered this solution of co-construction?

21. 248: Did the patients and healthcare professionals mention it was a good idea to learn a common reference system or do the authors make this claim based on literature?

22. 258: Who suggested to create a digital database? Seems like a pretty big leap from a conversation about the use of visual aids, and most patients not having much experience with it as the authors describe, to them suggesting the development of a digital database.

23. 265-266: “We have senses… been explicitly stated”. This is not data but the authors’ interpretation.

24. 295-296: Could the authors please elaborate on how their experience has helped to mitigate bias in the research

25. 298: If the authors were interested to explore stakeholders’ perceptions on potential uses of visual aids, the statement regarding generalisation might be true to some extent. But findings on how visual aids are being used in a single location are not necessarily generalisable to different healthcare context. The authors should be more realistic and specific about the extent to which their findings are generalisable.

Discussion

26. Although there appear to be some interesting findings, this section is confused as well. It presents some data that is not apparent in the results section, e.g. lines 332-33: “several healthcare professionals… for translation purposes” and lines 343-345: “the healthcare professionals… context of the patient.”

27. 307: While I see how the classification of visual aids in practice can be helpful for the research team, I am not convinced that this is a priority finding in the context of use of visual aids and barriers and facilitators in practice. Perhaps the authors can restructure the discussion to place more emphasis on discussing the research questions.

28. 302-303: These sentences seem to contradict each other: “they do not… by their graphic characteristics” vs “…emphasize the importance of aesthetics”, which is shaped by the graphic characteristics.

Conclusion

29. 356: The conclusion does not answer the research question of how visual aids are used in ambulatory practice and identifying practical constraints. Instead, the conclusions mainly focus on integrating visual aids in clinical practice, although it does seem to address the question on benefits of visual aids in practice.

30. 363-364: I am unsure what this conclusion is based on.

Practice implication

31. 366-368: This might be a helpful project, but does not follow from the described work and could have been proposed regardless of the specific study findings.

Others

32. Spelling/grammar could be improved in places, e.g. leave out ‘have’ in several instances (lines 34, 104, 113…)

33. I would suggest to make the title a more specific: single-site and ambulatory care

6. PLOS authors have the option to publish the peer review history of their article (what does this mean?). If published, this will include your full peer review and any attached files.

Reviewer #1: **Yes: **Sander D. Borgsteede

Reviewer #2: No

---

## [Author Response · Author response to Decision Letter 0]

11 May 2021

Thank you for your detailed reading and your helpful comments. We have given all attention to all the comments and answered all of them in the Revision letter.

---

## [Decision Letter · Decision Letter 1]

15 Jun 2021

PONE-D-20-29419R1

Visual aids in clinical practice: experiences, perceptions and needs of patients and healthcare professionals

PLOS ONE

Dear Dr. Hafner,

Thank you for submitting your manuscript to PLOS ONE. After careful consideration, we feel that it has merit but does not fully meet PLOS ONE’s publication criteria as it currently stands. Therefore, we invite you to submit a revised version of the manuscript that addresses the points raised during the review process.

We look forward to receiving your revised manuscript.

Kind regards,

Barbara Schouten

Academic Editor

PLOS ONE

Journal Requirements:

Additional Editor Comments (if provided):

One reviewer has now commented on the revised draft of your manuscript. As you well see, he has some remaining comments to further improve upon your manuscript. In particular, the point raised about whether or not ethical clearance has been obtained to perform your study should be clarified.

Reviewers' comments:

Reviewer's Responses to Questions

**Comments to the Author**

1. If the authors have adequately addressed your comments raised in a previous round of review and you feel that this manuscript is now acceptable for publication, you may indicate that here to bypass the “Comments to the Author” section, enter your conflict of interest statement in the “Confidential to Editor” section, and submit your "Accept" recommendation.

Reviewer #1: (No Response)

2. Is the manuscript technically sound, and do the data support the conclusions?

Reviewer #1: Partly

3. Has the statistical analysis been performed appropriately and rigorously? 

Reviewer #1: N/A

4. Have the authors made all data underlying the findings in their manuscript fully available?

Reviewer #1: Yes

5. Is the manuscript presented in an intelligible fashion and written in standard English?

Reviewer #1: Yes

6. Review Comments to the Author

Reviewer #1: Some minor remarks (line numbers from version with track changes):

Definition of VA (introduction): you seem to focus on non-moving images, also illustrated by your examples. Please add this to your definition.

Methods: include in the manuscript when the interviews were performed.

Approach: purposive selection: please include your criteria for selection of HP as well (including motivation). You should mention some key characteristics of these professionals (experience, gender) as well, and mention if the were distributed well over the focus groups

Ethics (p6, line 102-103): it is not clear if the proposal was submitted to the Research Ethics Council or not. The authors state: 'full submission was not required'. Does this mean that there was a procedure that the Counsil approved a brief version of the protocol? Or was the protocol not submitted nor judged in any respect at all?

Minor detail (p7, line 123) authors state that seperation of patients SEEMED important. It looks like this WAS CONSIDERED by the authors, and not based on a result of this study.

Results (p 17, line 287, semantic point): main characteristic of interactive VA was their context of use. I doubt if you can call this a characteristic, if this is part of your definition. It is valuable to read how HP and professionals interact with VA, but this is not be both an essential element of the definition and a characteristic.

P20, line 370: please include a citation for 'pateints fear time of interaction with HP may be suppressed' as this seems an important point that comes back in the summary of results.

In my opinion, the abstract does not cover your main findings. You might consider to include the concepts of stand alone and interactive VA, and the purpose of VAs in the abstract.

7. PLOS authors have the option to publish the peer review history of their article (what does this mean?). If published, this will include your full peer review and any attached files.

Reviewer #1: **Yes: **Sander D. Borgsteede

---

## [Author Response · Author response to Decision Letter 1]

9 Dec 2021

Thank you again for the helpful comments. We have answered all of the comments/questions in the attached rebutal letter (revision letter). 

We thank you for the delay that allowed us to work on the changes.

---

## [Editor Report · Decision Letter 2]

12 Jan 2022

Visual aids in clinical practice: experiences, perceptions and needs of patients and healthcare professionals

PONE-D-20-29419R2

Dear Dr. Hafner,

We’re pleased to inform you that your manuscript has been judged scientifically suitable for publication and will be formally accepted for publication once it meets all outstanding technical requirements.

Kind regards,

Barbara Schouten

Academic Editor

PLOS ONE

---

## [Editor Report · Acceptance letter]

24 Jan 2022

PONE-D-20-29419R2 

Visual aids in ambulatory clinical practice: experiences, perceptions and needs of patients and healthcare professionals 

Dear Dr. Hafner:

I'm pleased to inform you that your manuscript has been deemed suitable for publication in PLOS ONE. Congratulations! Your manuscript is now with our production department. 

Kind regards, 

on behalf of

Dr. Barbara Schouten 

Academic Editor

PLOS ONE